# Risk Model Validation: An Intraday VaR and ES Approach Using the Multiplicative Component GARCH

**Ravi Summinga-Sonagadu**  **and Jason Narsoo \***

Department of Economics and Statistics, University of Mauritius, Réduit 80837, Mauritius;
ravisonagadu@gmail.com
**\*** Correspondence: j.narsoo@uom.ac.mu; Tel.: +230-403-7948

**Abstract:** In this paper, we employ 99% intraday value-at-risk (VaR) and intraday expected shortfall (ES) as risk metrics to assess the competency of the Multiplicative Component Generalised Autoregressive Heteroskedasticity (MC-GARCH) models based on the 1-min EUR/USD exchange rate returns. Five distributional assumptions for the innovation process are used to analyse their effects on the modelling and forecasting performance. The high-frequency volatility models were validated in terms of in-sample fit based on various statistical and graphical tests. A more rigorous validation procedure involves testing the predictive power of the models. Therefore, three backtesting procedures were used for the VaR, namely, the Kupiec's test, a duration-based backtest, and an asymmetric VaR loss function. Similarly, three backtests were employed for the ES: a regression-based backtesting procedure, the Exceedance Residual backtest and the V-Tests. The validation results show that non-normal distributions are best suited for both model fitting and forecasting. The MC-GARCH(1,1) model under the Generalised Error Distribution (GED) innovation assumption gave the best fit to the intraday data and gave the best results for the ES forecasts. However, the asymmetric Skewed Student's-t distribution for the innovation process provided the best results for the VaR forecasts. This paper presents the results of the first empirical study (to the best of the authors' knowledge) in: (1) forecasting the intraday Expected Shortfall (ES) under different distributional assumptions for the MC-GARCH model; (2) assessing the MC-GARCH model under the Generalised Error Distribution (GED) innovation; (3) evaluating and ranking the VaR predictability of the MC-GARCH models using an asymmetric loss function.

**Keywords:** model validation; high-frequency; Multiplicative Component Generalised Autoregressive Heteroskedasticity (MC-GARCH); error distributions; intraday value-at-risk (VaR); intraday expected shortfall (ES); backtests

## 1. Introduction

Since the financial crisis of 2008, there has been an ever-growing need for financial entities to accurately assess their exposure to financial risks. Risk being commonly characterised by increasing volatility in the financial market, the modelling and forecasting of the volatility have become a very important research area among academics and practitioners in the last decade. According to Poon and Granger (2001), volatility can be viewed as a 'barometer for the vulnerability of financial markets and the economy'. It is, therefore, important to forecast volatility accurately. Volatility is also an essential tool in the computation of other risk metrics such as Value-at-Risk (VaR) and Expected Shortfall (ES).

Value-at-Risk (VaR) is a mandatory risk management tool in the insurance and banking industry as per the regulatory norms of the Solvency II framework (Solvency II European Directive (2009/138/EC))

and the Basel committee (BCBS 2010), respectively. However, it was observed that under the stress conditions of the global financial crisis, VaR forecasts were exceeded multiple times. In the Basel Committee on Banking Supervision BCBS (2016) report, it was concluded that during times of significant financial market stress, ES will ensure that tail risk and capital adequacy are captured in a more prudent manner. Interest in ES grew alarmingly ever since the Basel Committee on Banking Supervision (BCBS) brought forward their intention to replace VaR with ES, BCBS (2012).

For a long time, researchers and academics have made use of low-frequency data in financial time series analysis to forecast risk metrics such as volatility, Value-at-Risk (VaR) and Expected Shortfall (ES). However, low-frequency data misses out on precious information and as addressed by Engle and Russell (2004): "Like the view from the airplane above, classic asset pricing research assumes only that prices eventually reach their equilibrium value, the route taken and speed of achieving equilibrium is not specified". Low-frequency data lacks important details on the price adjustment compared to analysing high-frequency data. High-frequency data is defined as observations made over a short period of time, usually a day or less.

As mentioned by Zivot (2005), the unique characteristics of high-frequency data render the process of econometric and statistical analysis even more complicated. This in turn makes the forecasting of intraday VaR and ES quite challenging. For instance, econometric models or the modelling process should be able to take into account the intraday periodicity and the high excess kurtosis of the data to provide reliable forecast of the risk metrics. Moreover, the number of observations in high-frequency financial datasets can be overwhelming at times, and these observations may also be irregularly time-spaced.

Although forecasting VaR and ES using high-frequency data is challenging, it is also meaningful at the same time. As frequently mentioned in the literature, the volatility model is a fundamental ingredient which influences the measurement of both VaR and ES. It has been shown that the use of high-frequency data provides much more accurate estimates of volatility, Giot (2000). Due to the intense trading system nowadays, firms are forced to constantly build and devise strategies with the aim to beat the market. As mentioned by Müller (2000), it is no longer adequate to analyse these risk metrics based on daily data only. Today, more and more intraday price movements can be observed. Therefore, intraday VaR and ES estimates might be very beneficial to short-term traders involved in algorithmic and high-frequency trading, since the real-time market risk is quantified.

Despite the growing amount of research in the field of high-frequency financial data analysis, few studies have focused on model validation and high-frequency risk measures. This study contributes to the literature in the following ways:

(1)  A rigorous model validation, both in terms of in-sample fit and out-sample performance for the Multiplicative Component Generalised Autoregressive Heteroskedasticity (MC-GARCH) model under five error distributions is provided. Statistical and graphical tests are conducted to validate the models.

(2)  One component of the MC-GARCH model is the daily variance forecast. For this purpose, the GARCH(1,1) and EGARCH(1,1) under the five error distributions are compared and the best model among the 10 GARCH models is used to forecast the daily variance.

(3)  The modelling and forecasting performance of the MC-GARCH model under different distributional assumptions is assessed in this study.

(4)  The 99% intraday VaR is forecasted and three backtesting procedures are used. This is the first study to assess the VaR predictive ability of the MC-GARCH models by using an asymmetric VaR loss function.

(5)  This is the first study to forecast the intraday expected shortfall under different distributional assumptions for the MC-GARCH model. Again, three backtests are used including the recently proposed ES regression backtest of Bayer and Dimitriadis (2018).

Due to the high importance of risk management, the results of this study may contribute in many fields. This study is highly relevant to the banking industry since banks are required to calculate risk metrics on a daily basis for internal control purposes and for determining their capital requirements. Risk measurement is also essential to the insurance industry from the pricing of insurance contracts to determining the Solvency Capital Requirement (SCR), and therefore, the results of this study might be useful. Any other organisation with exposure to some kind of financial risk might benefit from this study. For instance, as mentioned by Culp et al. (1998), an airline company might use these intraday risk metrics to assess their exposure to jet fuel prices.

The rest of this paper is organized as follows. Section 2 provides a brief literature review on the MC-GARCH model, followed by Section 3, which details the various methodologies employed in this study. Section 4 presents the application of the MC-GARCH models and the various backtesting results. Finally, Section 5 will seal off the research with a summing up of the entire research outcome and will also provide recommendations for further study.

## 2. Past Studies on MC-GARCH Model

The literature on Autoregressive Conditional Heteroscedasticity (ARCH) models and Generalised Autoregressive Conditional Heteroscedasticity (GARCH) has grown impressively since they were first introduced by Engle (1982) and Bollerslev (1987), respectively. As noted in Andersen and Bollerslev (1997), since GARCH models are associated with a geometric decay in their autocorrelation structure of returns, they cannot take into account the pronounced intraday seasonal pattern present in the high-frequency financial returns. Over the years, to circumvent this limitation, researchers have come up with different solutions by augmenting the basic GARCH family of models. For instance, Andersen and Bollerslev (1997, 1998) and Andersen et al. (1999) took a novel approach by first deseasonalising the absolute returns prior to model fitting. The year 2011 saw the introduction of the MC-GARCH model of Engle and Sokalska (2011), which is a more sophisticated model designed specifically for high-frequency financial time series data. Basically, in this model, the variance part is decomposed into three multiplicative components: a daily component, a diurnal component and a stochastic volatility component. What makes the MC-GARCH model different from other typical GARCH models is that it includes a component which independently takes into account the intraday seasonality.

Previous studies have shown that, indeed, the MC-GARCH model is well capable of forecasting intraday volatility and risk metrics. The MC-GARCH model was applied to three equally spaced intervals of 1 min, 5 min and 10 min intraday data of Australia's S&P/ASX-50 stock market by Singh et al. (2013). The model yielded satisfactory results for intraday VaR forecast. Their results were supported by another study by Diao and Tong (2015), who found that the MC-GARCH model performed well in forecasting the intraday VaR in Chinese stock market. The dataset used was 5-min intraday returns of CSI7-300 index. In both studies, the innovation process of the variance equation was assumed to have a Gaussian distribution.

Narsoo (2016) applied the MC-GARCH model under four innovation distributions namely the Gaussian, the symmetric Student's-t, the skewed Student's-t and the reparametrised Johnson SU (JSU) distribution on the intraday 1-min EUR/USD exchange rates data to forecast the 99% VaR. Based on the Kupiec's test, it was concluded that the Skewed Student's-t MC-GARCH model delivered the best VaR forecast.

However, there are still a lot of open research areas on the MC-GARCH model. For instance, there is no study dealing with the model validation of the MC-GARCH model under various distributional assumptions and assessing the performance, both in terms of model fitting and forecasting. Also, there is no study on the expected shortfall (ES) forecasting performance of the MC-GARCH model under different error distributions. This paper therefore contributes to the high-frequency trading and backtesting literature by forecasting the intraday Value-at-Risk (VaR) and intraday Expected Shortfall (ES) at 99% confidence level using the MC-GARCH model under five distributional assumptions, which are the Normal, the Student's-t, the Skewed Student's-t distribution, the reparametrised Johnson SU

(JSU) and the Generalized Error Distribution (GED). After model fitting, the models will be validated in terms of in-sample fit based on a series of statistical and graphical tests. Due to the low statistical power of the Kupiec's test, two other backtests are also employed to rigorously assess the competency of the MC-GARCH models in predicting the intraday VaR. Three backtesting procedures will also be used to test the ES forecasting ability of the models.

## 3. Methodology

This study focuses on forecasting the intraday Value-at-Risk (VaR) and intraday Expected Shortfall (ES) at 99% confidence level using the MC-GARCH model under five distributional assumptions. This section explains the various models used to model both daily and intraday data. The backtesting procedures to assess the intraday VaR and ES forecasts are also presented.

### 3.1. Model Specification

#### 3.1.1. Models for the Daily Variance Component

#### GARCH(1,1)

The standard GARCH(1,1) model can be specified by the following set of equations:

$$r_t = m_t + \varepsilon_t$$

$$h_t = \omega + \alpha_1 \varepsilon_{t-1}^2 + \beta_1 h_{t-1}$$

where $m_t$ is the conditional mean process made up of both autoregressive (AR) and moving averages (MA) terms and $r_t$ represents the daily log returns. We assume $\varepsilon_t$ is the error term which can be decomposed as $\varepsilon_t = \sqrt{h_t z_t}$. The second equation is the variance equation and $h_t$ is the volatility process to be estimated. The innovation term, $z_t$ are i.i.d. variables.

In the variance equation, $\omega > 0$, $\alpha_1 > 0$, $\beta_1 > 0$ and $\alpha_1 + \beta_1 < 1$ to satisfy wide-sense stationarity.

#### EGARCH(1,1) Model

The Exponential GARCH model (EGARCH) of Nelson (1991) is also employed. It captures the asymmetric effects between positive and negative asset returns and models the logarithm of the conditional variance $h_t$. The EGARCH(1,1) specification has the following form:

$$\ln(h_t) = \omega + \frac{\alpha_1 \varepsilon_{t-1} + \gamma_1 |\varepsilon_{t-1}|}{h_{t-1}} + \beta_1 \ln(h_{t-1})$$

To ensure non-negative variance, the model is an AR(1) on $\ln(h_t)$ instead of $h_t$.

#### 3.1.2. Model for Intraday Returns

#### MC-GARCH(1,1) Model

The Multiplicative Component GARCH model (MC-GARCH) is a variant of the GARCH model which is specifically designed to model and forecast the intraday returns of financial assets. Basically, in this model, the conditional variance equation is specified by a multiplicative product of a daily volatility component, a diurnal volatility component and also a stochastic/intraday volatility component. For the sake of clarity, let $R_{t,i}$ be the conditional compounded return series for a particular financial asset $A$, where $t$ is representing any particular day and $i$ is the regularly spaced intraday time period. In the MC-GARCH model, the intraday return process of $R_{t,i}$ may be represented as follows:

$$R_{t,i} = \sqrt{h_t s_i q_{t,i}} \varepsilon_{t,i}$$

where $\varepsilon_{t,i} \sim N(0,1)$ and

- $h_t$ denotes the daily variance component
- $s_i$ denotes the diurnal/calendar variance component in each intraday period
- $q_{t,i}$ denotes the intraday variance component
- $\varepsilon_{t,i}$ is an error term following a specified distribution

This study employs GARCH and EGARCH to forecast the daily variance component $h_t$, based on the paper by Andersen and Bollerslev (1997). The choice of the model is based on the best-performing one among the GARCH and EGARCH models under five error distributions, which are the normal distribution, the Student's-t distribution, the Generalised Error Distribution (GED), the skewed Student's-t and the Johnson SU (JSU) distribution.

The diurnal volatility component $s_i$, is estimated as the variance of intraday returns in each regularly spaced intraday time period as represented below:

$$\frac{R_{t,i}^2}{h_t} = s_i q_{t,i} \varepsilon_{t,i}^2$$

$$s_i = \frac{1}{T} \sum_{t=1}^{T} \frac{R_{t,i}^2}{h_t}$$

By using the daily variance and the diurnal variance, the returns are normalized in the following way:

$$\begin{aligned} z_{t,i} &= \frac{R_{t,i}}{\sqrt{h_i s_i}} \\ &= \sqrt{q_{t,i}} \varepsilon_{t,i} \end{aligned}$$

After the normalization of the returns by both the daily and diurnal variance, the next step consists of modelling the stochastic intraday variance component $q_{t,i}$ as a GARCH(1,1) process, which is given as follows:

$$q_{t,i} = \omega^* + \alpha_1^* \left(\frac{R_{t,i-1}}{\sqrt{h_t s_{i-1}}}\right)^2 + \beta_1^* q_{t,i-1}$$

where $\omega^* > 0$, $\alpha_1^* \geq 0$, $\beta_1^* \geq 0$.

### 3.2. Parameter Estimation

In this paper, all the parameters of the various GARCH models employed will be estimated using maximum likelihood estimation (MLE), since it is the most popular method for estimating GARCH type models. Moreover, this method yields asymptotically efficient parameter estimates for the GARCH models.

### 3.3. Value-at-Risk and Expected Shortfall Evaluation

Value-at-Risk Evaluation:

According to McNeil et al. (2005), the Value-at-Risk (VaR) of a portfolio at time $t$ for a given confidence level $q \in (0,1)$ is given by the smallest number $x_q$ such that the loss at time $t+1$, which is denoted by $X_{t+1}$, will be less than $x_q$ with probability $q$:

$$\begin{aligned} VaR_q^t &= \inf\{x_q \in \Re : P(X_{t+1} \leq x_q) \geq q\} \\ &= \inf\{x_q \in \Re : P(X_{t+1} > x_q) \leq 1 - q\} \end{aligned}$$

The one-step-ahead *VaR* is computed as follows:

$$VaR_{t+1}^\alpha = \mu_{t+1} + \sigma_{t+1} F^{-1}(\alpha)$$

where the probability distribution function *F* of the return innovations $z_t$, is strictly monotone or has a generalised inverse of the cumulative distribution function. In this paper, $z_t$ is assumed to follow five probability distributions namely the Normal, Student's-t, Skewed Student's-t, JSU and GED.

Expected Shortfall Evaluation:

The Expected Shortfall (ES) at a given level $\alpha$ is defined as being the expected value at time *t* of $X_{t+1}$, which is the loss in the next period conditional on the loss exceeding $VaR_\alpha^t$:

$$ES_\alpha^t = E_t[X_{t+1}|X_{t+1} < VaR_\alpha]$$

$$ES_\alpha = \frac{1}{1-\alpha} \int_\alpha^1 VaR_x \, dx$$

According to García Jorcano (2018), the one-step-ahead ES can be further simplified using the properties of the expectation operator:

$$ES_{t+1}^\alpha = \mu_{t+1} + \sigma_{t+1}\mathbb{E}_{t+1}[z_{t+1}|z_{t+1} < F^{-1}(\alpha)]$$

where:

- $z_{t+1} = \frac{X_{t+1}-\mu_{t+1}}{\sigma_{t+1}}$
- $F^{-1}(\alpha) = \frac{VaR_{t+1}^\alpha - \mu_{t+1}}{\sigma_{t+1}}$

### 3.4. Backtesting

After forecasting the risk metrics VaR and ES, a backtesting procedure is employed to assess the accuracy of the forecasts. In the backtesting procedure, actual profits and losses are compared to the estimates of VaR and ES in a systematic manner.

3.4.1. Value-at-Risk Backtesting

**Kupiec's Unconditional Coverage Test**

The Kupiec's test was developed by Kupiec (1995) and is the most famous VaR test that is based on failure rates. It is also known as the proportion of failures (POF) test. The null hypothesis of the test assumes that the number of exceptions follows a binomial distribution.

The null hypothesis for the test is as follows:

$$H_0 = p = \hat{p} = \frac{x}{T}$$

where *T* is the number of observations and *x* is the number of exceptions.

The test is in fact a likelihood ratio test where the test statistics are as follows:

$$LR_{POF} = -2\ln\left(\frac{(1-p)^{T-x}p^x}{[1-(x/T)]^{T-x}(x/T)^x}\right)$$

Under the null hypothesis, the $LR_{POF}$ is asymptotically chi-square distributed with one degree of freedom.

**A Duration-Based Approach to VaR Backtesting**

According to Christoffersen and Pelletier (2003), a more robust test to determine the adequacy of a risk model is by considering the duration between VaR violations. Ideally, the duration between the VaR violations should be independent of one another and should not cluster. The null hypothesis of

this test is that under a correctly specified risk model, the VaR violations should be memoryless and should therefore follow an exponential distribution as follows:

$$g(d_v; \alpha) = \alpha \exp(-\alpha d_v)$$

Under the alternative hypothesis, a Weibull distribution is used for the duration variable, since it embeds the exponential distribution as a restricted case:

$$h(d_v; a, b) = a^b b d_v^{b-1} \exp[-(a d_v)^b]$$

Also, $H_{0,IND} : b = 1$ and $H_{0,CC} : b = 1, a = \alpha$, where IND denotes independence and CC denotes Conditional Coverage.

**Asymmetric VaR Loss Function**

Even though the two VaR backtesting procedures discussed above are highly relevant for testing the model adequacy, they do, however, fail to judge the model based on its predictive accuracy. In other words, they do not provide statistical evidence as to whether there is any difference in the forecasting performance between the different models employed. Therefore, González-Rivera et al. (2004) proposed an asymmetric VaR loss function to compare the performance of the different model on the basis of the loss function. The loss function is defined as:

$$\updownarrow(r_{t+1}, VaR_{j,t+1|t}^{\tau}) = T_0^{-1} \rho_{\tau}(r_{t+1} - VaR_{j,t+1|t}^{\tau}), t = 1, 2, \ldots, T_0$$

where $T_0$ is the length of the backtesting period, $j$ is the model indicator, $r_{t+1}$ denotes the return at time $t + 1$, $VaR_{j,t+1|t}^{\tau}$ denotes the VaR at $t + 1$ given the information set up to time $t$. Moreover, $\rho_{\tau} = z(\tau - I_{-\infty,0}(z))$ denotes the $\tau^{th}$ quantile loss function. Since it is an asymmetric loss function, it penalises observations below the $\tau^{th}$ quantile level more heavily as compared to observations above it. The best model is the one which minimises this loss function.

**Model Confidence Set Procedure**

Hansen et al. (2011) proposed the model confidence set (MCS) procedure, whereby a sequence of statistical tests are carried out with the objective of building a "Superior Set of Models" (SSM). Basically, the equal predictive ability (EPA) test statistic is calculated for an arbitrary loss function satisfying the general weak stationarity conditions. In this procedure, the loss function employed is the asymmetric VaR loss function of González-Rivera et al. (2004). For a chosen level of confidence, the null hypothesis stating EPA is not rejected. This procedure is implemented to rank the models, in ascending order, according to their VaR forecasting power.

3.4.2. Expected Shortfall Backtesting

**The Bivariate ES Regression Backtest**

The first ES backtest that will be considered is the very recently proposed Bivariate ES Regression Backtest of Bayer and Dimitriadis (2018). They proved that this backtest has far more power than other ES backtests. It is also more convenient for regulators since it is the only backtest method in the literature which uses only the ES forecasts for the backtesting of the risk metric.

The Bivariate ES Regression Backtest simply tests if a series of ES forecasts denoted by $\{\hat{e}_t, t = 1, \ldots T\}$ from a forecasting model is specified correctly with respect to the series of realized returns denoted by $\{Y_t = 1, \ldots, T\}$. Basically, in this backtest, the returns $Y_t$ are regressed on the ES forecasts $\hat{e}_t$ including an intercept term which is designed particularly for the functional ES.

$$Y_t = \alpha + \beta \hat{e}_t + u_t^e \tag{1}$$

where $ES_\tau(u_t^e|\mathcal{F}_{t-1}) = 0$. Moreover, the condition on the error term can be specified in another way since $\hat{e}_t$ is generated using the same filtration set $\mathcal{F}_{t-1}$:

$$ES_\tau(Y_t|\mathcal{F}_{t-1}) = \alpha + \beta\hat{e}_t$$

The null hypothesis $H_0$ is tested against the alternative hypothesis $H_1$ where:

$$H_0 : (\alpha, \beta) = (0, 1)$$

$$H_1 : (\alpha, \beta) \neq (0, 1)$$

The null hypothesis $H_0$ states that the ES forecasts are specified correctly since $\hat{e}_t = ES_\tau(Y_t|\mathcal{F}_{t-1})$. This backtest is called a bivariate backtest, since the parameters $\alpha$ and $\beta$ are tested simultaneously based on the regression framework.

The estimation of Equation (1) is carried out by the semiparametric estimation of the joint system:

$$Y_t = \gamma + \delta\hat{e}_t + u_t^q$$

$$Y_t = \alpha + \beta\hat{e}_t + u_t^e$$

where $\mathcal{Q}_\tau(u_t^q|\mathcal{F}_{t-1}) = 0$ and $ES_\tau(u_t^e|\mathcal{F}_{t-1}) = 0$. Thus, $Y_t$ is the response variable and $(1, \hat{e}_t)$ are the explanatory variables in the regression. A Wald statistic is computed incorporating the parameters $(\alpha, \beta)$ to test the null hypothesis as follows:

$$T_{ESR} = ((\hat{\alpha}, \hat{\beta})' - (0, 1)')'\hat{\Sigma}_{ES}^{-1}((\hat{\alpha}, \hat{\beta})' - (0, 1)')'$$

where $\hat{\Sigma}_{ES}$ is an estimator for the covariance matrix of the M-estimator for the parameters $\alpha$ and $\beta$. The test statistic follows a chi-square distribution with two degrees of freedom.

Bayer and Dimitriadis (2018) also showed that the backtest procedure has even greater power when combined with bootstrapping. The backtest will therefore also be carried out using bootstrapping where B = 1000 bootstrap Wald statistics will be computed. The bootstrap *p*-value will simply be the share of the 1000 bootstrap test statistics greater or equal to the test statistic for the original sample.

**Exceedance Residual (ER) Backtest**

McNeil and Frey (2000) was among the first to propose an expected shortfall backtesting procedure. This procedure analyses the difference between the next period's return $X_{t+1}$ and $ES_q^t(X_{t+1})$ which is the expected shortfall at time $t$, conditional on the fact that $X_{t+1}$ exceeds the VaR at time $t$, $VaR_q^t(X_{t+1})$.

$$r_{t+1} = \frac{x_{t+1} - \hat{E}S_q^t(X_{t+1})}{\hat{\sigma}_{t+1}}$$

Under the null hypothesis ($H_0$), they postulated that the modified series $r_t$ should be i.i.d with mean 0 and variance 1. To test $H_0$, the non-parametric bootstrapping method is employed on the $n$ observations in $r_t$ against the alternative hypothesis which states that the mean of excess violation of VaR is greater than 0. The bootstrap methodology was devised by Efron and Tibshirani (1994).

**V-Test for the Expected Shortfall**

Different methods to evaluate the performance of the ES estimates were proposed by McNeil et al. (2005). These methods were based on the relative size of test statistics. These test statistics are regarded more as a diagnostic tool than a formal statistical test, since there is no null hypothesis testing involved.

The first statistic, $V_1$, takes the average between the forecasted ES and the actual return whenever a VaR violation occurs. A correctly specified model should yield a value close to 0 for $V_1$. For a given probability $q$, $V_1$ is defined as:

$$V_1 = \frac{\sum_{t=1}^{T}(x_{t+1} - \hat{ES}_q^t(X_{t+1}))1_{\{X_{t+1}>\hat{x}_q^t\}}}{\sum_{t=1}^{T}1_{\{X_{t+1}>\hat{x}_q^t\}}}$$

where $T$ denotes the total number of ES estimates.

However, the main drawback of $V_1$ is that it is too dependent on the VaR estimates. Therefore, McNeil et al. (2005) proposed a second test statistic $V_2$, which is defined as follows:

$$V_2 = \frac{\sum_{t=1}^{T}(x_{t+1} - \hat{ES}_q^t(X_{t+1}))1_{\{D_t>D_q\}}}{\sum_{t=1}^{T}1_{\{D_t>D_q\}}}$$

$D_t = (x_{t+1} - \hat{ES}_q^t(X_{t+1}))$ and $D_q$ is the empirical q-quantile of $\{D_t, t = 1, 2, \ldots, T\}$.

A third measure was also brought forward which combines $V_1$ and $V_2$ to strike a balance between the test statistic $V_1$, which relies too heavily on theory, and the test statistic $V_2$, which is more practically oriented. This measure is denoted by $V$, and it is defined as:

$$V = \frac{|V_1| + |V_2|}{2}$$

A good model would therefore bring the test statistics $V_2$ and $V$ close to 0.

## 4. Estimation Results

### 4.1. Data Description

The intraday 1-min EUR/USD exchange rate price dataset consists of 28,290 observations for the month of February 2016 equivalent to 21 days intraday logarithmic returns. Moreover, the daily returns of the EUR/USD exchange rate are also used since a GARCH model will be employed to forecast the daily variance component in the MC-GARCH model. The daily data of the EUR/USD exchange rate prices span from 2 December 2003 to 29 February 2016 and consists of 3160 observations. The intraday dataset is split into two samples, where a sample of 20 days is used for estimating the models and a sample of 1 day is used to assess the forecasting ability of the models.

The daily log return $r_t$ can be calculated as below. The same principle applies to the intraday log return process $R_{t,i}$.

$$r_t = \ln\left(\frac{P_t}{P_{t-1}}\right)$$

In the above equation, $P_t$ is the exchange rate price at time $t$ and $P_{t-1}$ is the exchange rate price at time $t-1$. Calculating the log returns actually transforms the financial time series into a stationary series. The Augmented Dickey-Fuller (ADF) test results presented in Appendix A actually confirm the stationary property of both the 1-min and daily exchange rate log-returns.

### 4.2. Heteroskedasticity and Normality Tests of the Return Series

Figure 1 shows the return series plot for the 1-min EUR/USD exchange rate returns. Figure 2 is the correlogram of the absolute returns for the 1-min EUR/USD returns for the month of February 2016. Clearly, a strong pattern repeating approximately every 1500 observations, corresponding to a day, can be observed. Volatility is high at the opening and closing hours. This depicts the strong intraday seasonality revealed in the high-frequency literature.

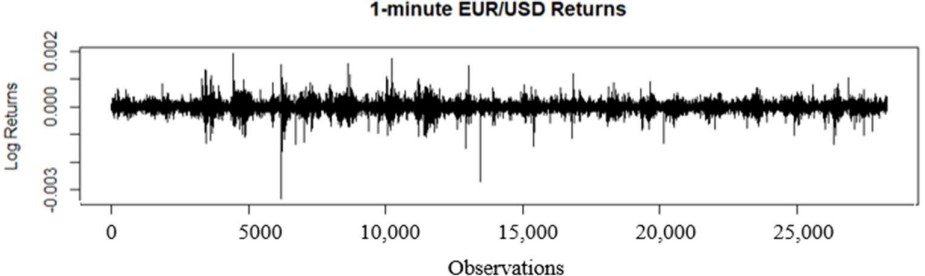

**Figure 1.** Return series plot for the 1-min EUR/USD exchange rate returns.

**Figure 2.** Correlogram of the absolute returns for the 1-min EUR/USD returns. ACF: Auto Correlation Function.

The descriptive statistics and normality tests of the EUR/USD exchange rate returns for the high-frequency 1-min returns and for the daily returns are presented in Appendix A.

*4.3. Identifying the Conditional Mean Equation*

The first step to the implementation of GARCH-type models for the conditional variance, involves identifying a suitable model for the conditional mean of the data. Literature suggests the implementation of an Auto Regressive Integrated Moving Average (ARIMA) model for modelling the conditional mean. Since both return series are stationary, the order of the parameter $d$ in the ARIMA$(p, d, q)$ model is equal to 0.

The next step consists of determining the order of the parameters $p$ and $q$ for the two return datasets. A graphical analysis of the Auto Correlation Function (ACF) and Partial Auto Correlation Function (PACF) of the two returns series is first carried out to visually determine the orders of their Auto Regressive Moving Average or ARMA$(p,q)$ model. The ACF for both returns series are plotted in Figure 3 along with their respective PACF.

While analysing the ACF and PACF plots, it seems that an ARMA(0,0) model is appropriate for both the 1 min returns and for the daily returns. To further confirm the order of the mean equation, several ARMA$(p,q)$ models are estimated, and the best model was chosen based on two criteria: the minimum Akaike information criterion (AIC) value and the maximum log-likelihood value. As stated by Mondal et al. (2014), the Box-Jenkins methodology states that the value of $p$ and $q$ for an ARIMA$(p, d, q)$ model should be equal to or less than 2, or the total number of parameters should be no more than 3. Therefore, the AIC and log-likelihood values are checked only for those ARMA model with parameters p and q having a value of 2 or less. The ARMA(0,0) model provided the lowest AIC value and the maximum log-likelihood value for both the 1-min return and for the daily returns series and therefore outperforms the other ARMA specifications for the conditional mean.

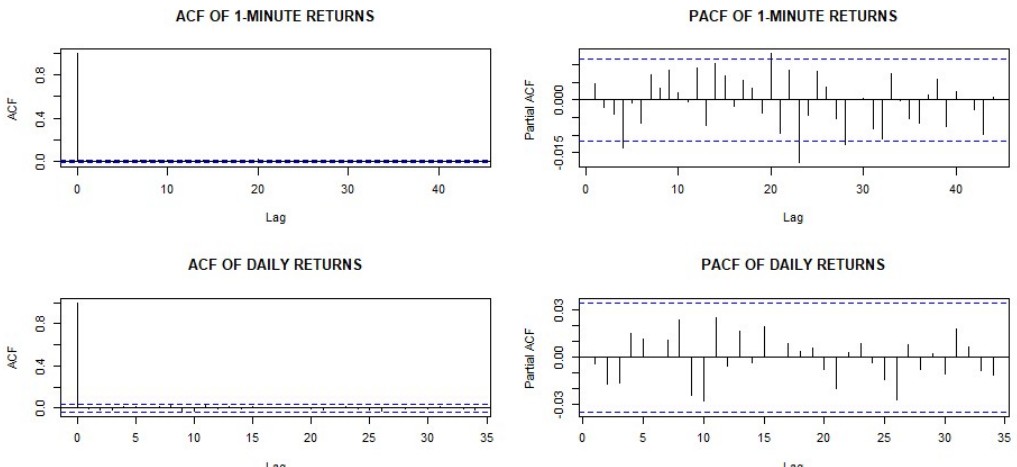

**Figure 3.** ACF and PACF plots for intraday and daily return series. PACF: Partial Auto Correlation Function.

### 4.4. Model Checking for the Mean Equation

According to Tsay (2005), there is a need to eliminate any significant correlations in the return series prior to fitting any GARCH-type model. The residuals of the mean equation are therefore tested for the presence of autocorrelations using the Ljung-Box Q test. All the *p* values were greater than 5% at 10 and 20 degrees of freedom, implying that the residuals of the mean equation are not serially autocorrelated for the two return datasets.

At this stage, since the two return datasets exhibit stylized features such as excess kurtosis and clustering of volatility and given the adequacy of the ARMA specifications for the mean equations, the specification of GARCH models to the returns datasets is analysed.

### 4.5. Estimation of Daily Variance Forecast

As stated by Engle and Sokalska (2011), the implementation of the MC-GARCH model first necessitates a model for the daily variance component. The GARCH(1,1) and the EGARCH(1,1) models are implemented under the five error distributions, and the best model is retained for the daily variance component. The parameter estimates of the GARCH-type models for the daily variance forecast are statistically significant. Since the parameter $\gamma_1$, which is the indicator for asymmetric volatility, was significant across all innovations for the EGARCH(1,1) model, this is indicative that an asymmetric GARCH might be preferred over a symmetric GARCH model. The parameter $\gamma_1$ being positive irrespective of the error distribution used, imply that shocks including both good news and bad news which may impact the daily EUR/USD returns will affect volatility for a long period of time in the future.

To choose the best model for the daily variance component, three criteria will be used: the AIC value, the Bayesian information criterion (BIC) value, and the log-likelihood value. The best model will be the one minimising both the AIC and BIC score while maximising the log-likelihood value. The results are presented in Tables 1 and 2.

**Table 1.** Daily variance forecast: GARCH(1,1) model.

| | GARCH(1,1) | | | | |
| --- | --- | --- | --- | --- | --- |
| | **Normal** | **Student's-t** | **Skewed Student's-t** | **JSU** | **GED** |
| AIC | −7.4566 | −7.4665 | −7.4659 | −7.4662 | −7.4704 |
| BIC | −7.449 | −7.4569 | −7.4543 | −7.4547 | −7.4608 |
| Log-likelihood | 11,781.8 | 11,798.33 | 11,798.32 | 11,798.9 | 11,804.5 |

**Table 2.** Daily variance forecast: EGARCH(1,1) model.

| | EGARCH(1,1) | | | | |
| --- | --- | --- | --- | --- | --- |
| | **Normal** | **Student's-t** | **Skewed Student's-t** | **JSU** | **GED** |
| AIC | −7.4602 | −7.4695 | −7.4689 | −7.4693 | −7.4734 |
| BIC | −7.4506 | −7.458 | −7.4555 | −7.4559 | −7.4619 |
| Log-likelihood | 11,788.4 | 11,804.14 | 11,804.15 | 11,804.8 | 11,810.2 |

It can be observed that the asymmetric EGARCH(1,1) model outperforms the GARCH(1,1) model under all error distributions since the former model yields the minimum AIC and BIC scores and yields higher log-likelihood values. This can be explained by the fact that the EGARCH(1,1) models are able to capture the leverage effect feature of the daily return series. However, the best-performing model is clearly the EGARCH model under the GED innovation assumption (EGARCH-GED) since this model yields the minimum AIC and BIC value while maximising log-likelihood. Hence, this model specification will be used for the daily variance forecast.

### 4.6. Fitting Performance

The MC-GARCH models is now fitted to the complete dataset of 28,289 1-min EUR/USD observations. Table 3 displays the results of the MC-GARCH parameter estimation. The corresponding *p*-values provided within parentheses.

**Table 3.** MC-GARCH(1,1) parameter estimates.

| | MC-GARCH(1,1) | | | | |
| --- | --- | --- | --- | --- | --- |
| | **Normal** | **Student's-t** | **Skewed Student's-t** | **JSU** | **GED** |
| $\mu$ | 0 (0.62316) | 0 (0.94151) | 0 (0.53047) | 0 (0.59145) | 0 (0.98539) |
| $\omega$ | 0.011999 (0) | 0.008613 (0) | 0.008651 (0) | 0.008727 (0) | 0.009911 (0) |
| $\alpha_1$ | 0.037484 (0) | 0.043874 (0) | 0.043774 (0) | 0.043762 (0) | 0.041275 (0) |
| $\beta_1$ | 0.950441 (0) | 0.949255 (0) | 0.949335 (0) | 0.949197 (0) | 0.949529 (0) |
| shape, $\nu$ | - | 6.893944 (0) | 6.894106 (0) | 1.878735 (0) | 1.340094 (0) |
| skewness | - | - | 1.012434 (0) | 0.037765 (0) | - |

All the parameter estimates are statistically significant at 5% level except for the conditional mean, which is insignificant at 5% level across all innovations for the MC-GARCH model, and also, the skewness parameter is insignificant for the JSU innovation. Almost all the parameter estimates being statistically significant gives an indication that the MC-GARCH models are correctly specified.

The statistical significance of the ARCH parameter $\alpha_1$ and GARCH parameter $\beta_1$ for all innovations of the MC-GARCH model suggests that lagged conditional variance and lagged squared disturbance have an impact on the current conditional variance. This simply implies that news about volatility from the previous periods have an explanatory power on the current volatility. Moreover, the high significance of the parameter $\alpha_1$ validates the presence of volatility clustering in the dataset.

The shape, $\nu$, parameter being highly statistically significant and greater than 4 for the Student's-t and skewed Student's-t error distributions and less than 2 for the GED innovation confirms the presence of thick tails as was shown by the excess kurtosis in the return dataset of the 1-min EUR/USD returns.

Moreover, the skewness parameter for the skewed Student's-t innovation being highly statistically significant also confirms the presence of skewness in the return series as was shown by the negative skewness of the dataset. These results suggest that a non-normal innovation might be a more suitable candidate for the MC-GARCH model.

To determine the best fitting model, three criteria will be used namely the AIC value, BIC value and the Log-Likelihood. These results are displayed in Table 4, below.

**Table 4.** Model selection for the MC-GARCH(1,1) model.

| | | | MC-GARCH(1,1) | | |
|---|---|---|---|---|---|
| | **Normal** | **Student's-t** | **Skewed Student's-t** | **JSU** | **GED** |
| AIC | −15.021 | −15.046 | −15.046 | −15.048 | −15.057 |
| BIC | −15.019 | −15.045 | −15.044 | −15.046 | −15.055 |
| Log-Likelihood | 212,463 | 212,826 | 212,827.4 | 212,849.3 | 212,976.5 |
| Rank | 5 | 4 | 3 | 2 | 1 |

From Table 4, it can be observed that the model yielding the worst results is the MC-GARCH model under the normal innovation. This can be explained by the fact that being a symmetric distribution and having a kurtosis of 3, the MC-GARCH model under the normal error distribution fails to capture features such as the leptokurtic nature of the 1-min EUR/USD returns.

The best model is clearly the MC-GARCH model under the GED innovation, since it yields the highest log-likelihood value of 212,976.5 while simultaneously yielding the lowest AIC value of −15.057 and BIC value of −15.055.

**Model Validation: In-Sample Fit:**

In this section, the chosen GARCH model is validated. The estimated, standardised residuals of the MC-GARCH model under the GED innovation should be independent and identically distributed and for this purpose, the ACF of the standardised residuals is analysed. It can be observed from Figure 4 that there are no significant lags, and therefore the residuals are not serially correlated and behave as a white noise process.

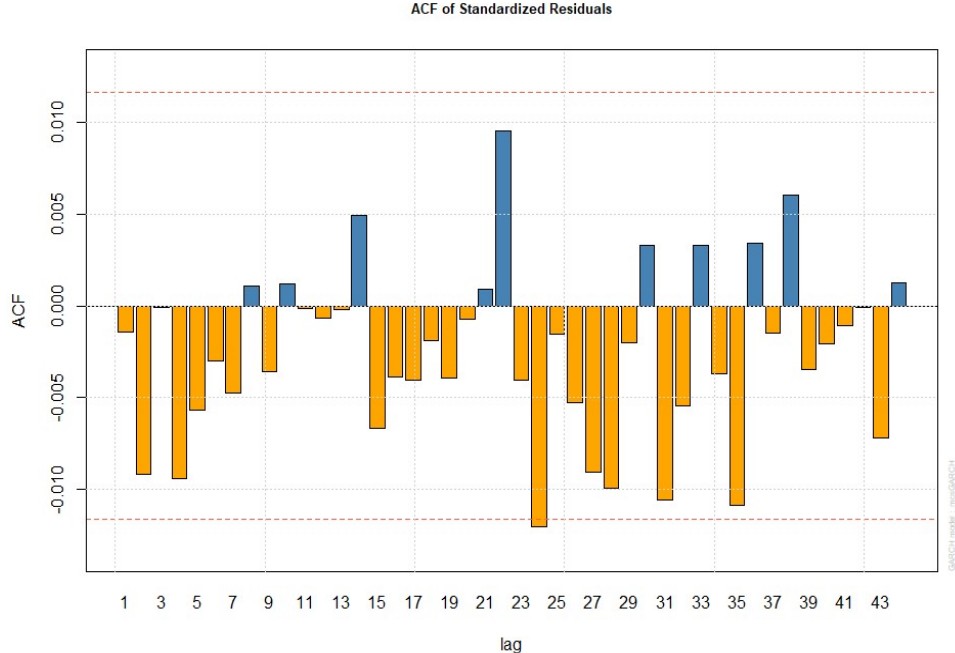

**Figure 4.** ACF of the standardised residuals.

The ARCH LM test was performed on the residuals of the MC-GARCH models at various lag lengths, and it was found that the null hypothesis stating that there is no ARCH effects cannot be rejected. This suggests that the conditional heteroskedasticity that was present in the raw series was successfully removed, thereby validating the MC-GARCH model. This result was backed by the Ljung-Box Test on the residuals.

The empirical density of the standardised residuals is plotted below to check whether the GED distribution gives the best fit.

Indeed, from Figure 5, it can be seen that the GED assumption fits well to the residuals, as compared to the other distributions.

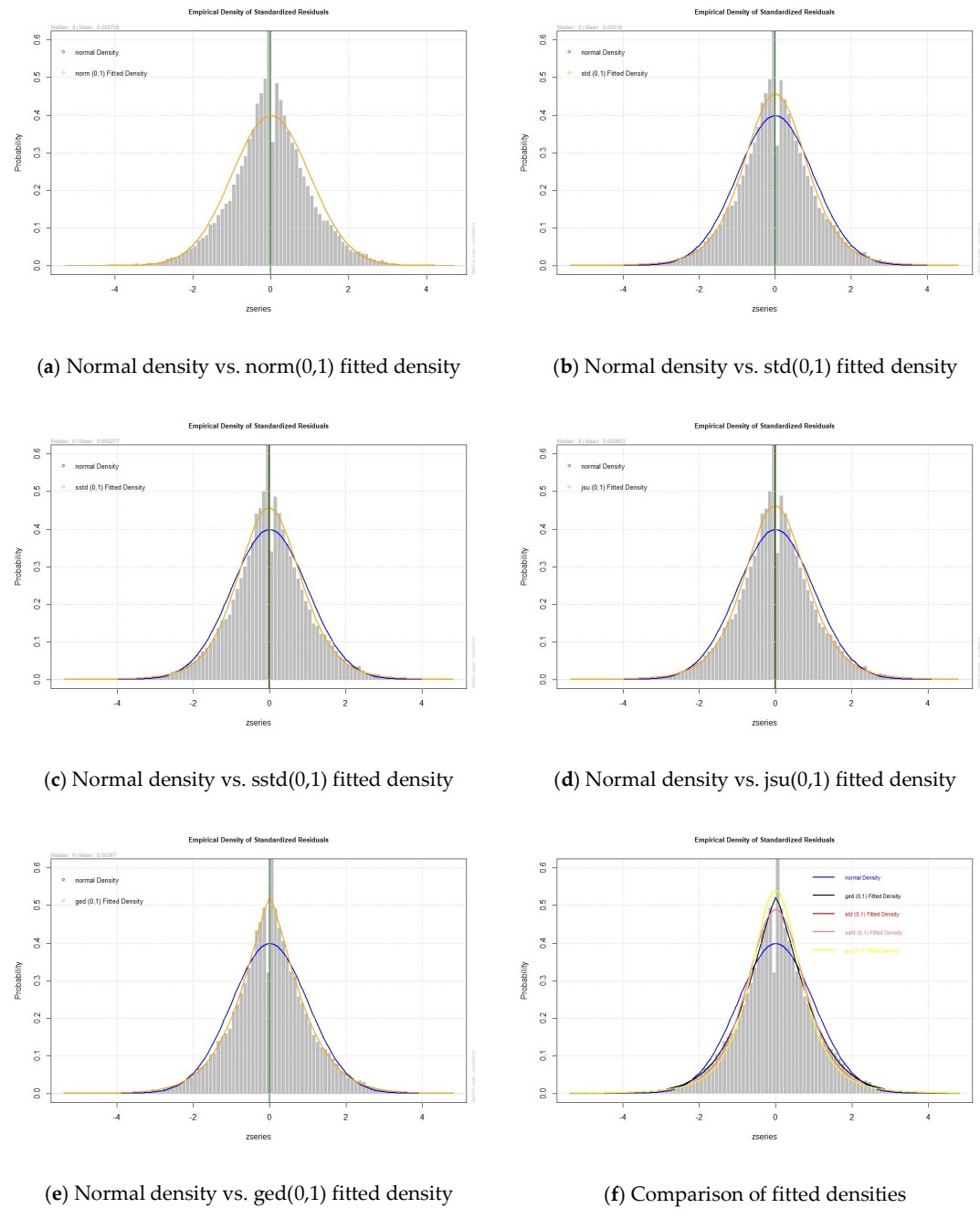

(**a**) Normal density vs. norm(0,1) fitted density

(**b**) Normal density vs. std(0,1) fitted density

(**c**) Normal density vs. sstd(0,1) fitted density

(**d**) Normal density vs. jsu(0,1) fitted density

(**e**) Normal density vs. ged(0,1) fitted density

(**f**) Comparison of fitted densities

**Figure 5.** Empirical density of the standardised residuals.

The figure below (Figure 6) displays the volatility decomposition into the different components for the MC-GARCH model under GED innovation: the diurnal component, the daily volatility component,

the intradaily volatility component, and the total composite volatility for the intraday 1-min data, which is obtained by combining the three volatility components.

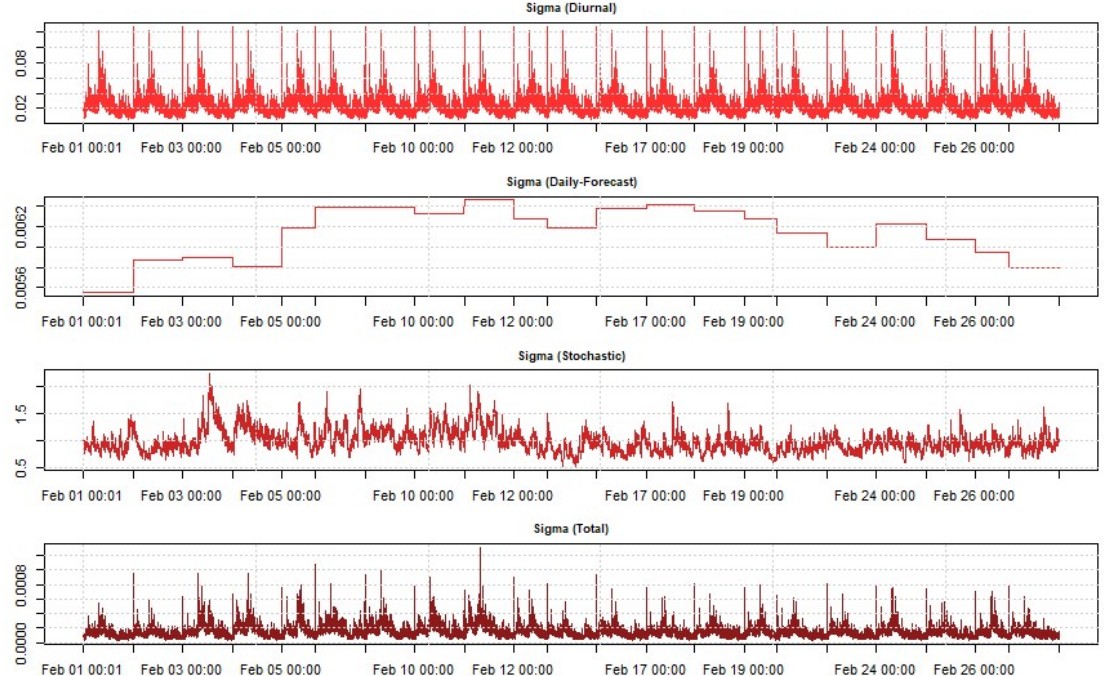

**Figure 6.** Diurnal component, daily volatility component, intradaily volatility component, and total composite volatility.

The MC-GARCH model is also valid, since it includes a component to cater for the intraday seasonality (sigma (Diurnal)).

### 4.7. Intraday VaR Forecast

The 99% intraday VaR is forecasted using the MC-GARCH models on the 1-min intraday return series. A rolling backtest procedure is then undertaken on the out-sample period and a moving window of 1 day will be used in the VaR backtesting procedure. The backtesting period is one day, which relates to 1500 1-min datapoints.

### 4.7.1. Kupiec's Test

The first backtest used is the Kupiec's unconditional coverage test, where the 1500 intraday VaR forecasts estimated are compared against the actual intraday returns. The results (Table 5) of the backtest speaks in favour of the MC-GARCH model, as all the models, except the MC-GARCH under the normal distribution, passed this test since the *p*-values, being greater than the 5% significance level, indicate that the null hypothesis cannot be rejected.

**Table 5.** Intraday VaR forecast: Kupiec's test.

|  | Normal | Student's-t | Skewed Student's-t | JSU | GED |
|---|---|---|---|---|---|
| Expected VaR Exceedances | 15 | 15 | 15 | 15 | 15 |
| Actual VaR Exceedances | 27 | 21 | 22 | 21 | 20 |
| Actual % | 1.80% | 1.40% | 1.50% | 1.40% | 1.30% |
| *p*-value | 0.005 | 0.142 | 0.089 | 0.142 | 0.217 |

4.7.2. VaR Duration Test

The results of the VaR duration test are displayed in Table 6.

**Table 6.** Intraday VaR forecast: duration-based approach VaR backtesting results.

| Model | $b$ | $p$-Value |
|---|---|---|
| MC-GARCH_norm | 0.877439 | 0.397975 |
| MC-GARCH_std | 0.85151 | 0.392917 |
| MC-GARCH_sstd | 0.85151 | 0.392917 |
| MC-GARCH_jsu | 0.85151 | 0.392917 |
| MC-GARCH_ged | 0.85151 | 0.392917 |

The second column '$b$' is the estimated Weibull parameter for the different models. Since the $p$-values for all models are greater than the significance level of 5%, this gives evidence that the duration of time between the VaR violations possess no memory and that they do not cluster. All the models passed the VaR duration-based backtest.

4.7.3. Backtesting VaR Using an Asymmetric Loss Function

A more rigorous backtesting procedure is carried out. As stated in Bernardi et al. (2014), though the Kupiec's test is able to compare VaR violations of several competing models, it fails, however, to rank the models according to their predictive accuracy of the VaRs. Moreover, many models satisfy the unconditional coverage test, as it is observed in this study. The risk manager therefore cannot select a unique method. Lopez (1998) suggested to measure the accuracy of VaR forecasts based on a loss function and the models are ranked accordingly.

To present results which are less sensitive to the low number of theoretical violations and to deal with the problem of the Kupiec's test, the Model Confidence Set (MCS) procedure proposed by Hansen et al. (2011) is applied together with the asymmetric VaR function of González-Rivera et al. (2004). The results for the MCS procedure are presented in Table 7. Only those models which passed the Kupiec's test and the VaR duration test are considered. The best-performing model according to this procedure is the MC-GARCH under the skewed Student's-t distribution, since it minimises the loss function.

**Table 7.** Intraday VaR forecast: MCS results and ranking.

| Superior Set of Model | | |
|---|---|---|
| Model | Rank | Loss ($\times 10^{-6}$) |
| MC-GARCH_std | 2 | 4.61995 |
| MC-GARCH_sstd | 1 | 4.615442 |
| MC-GARCH_jsu | 4 | 4.744222 |
| MC-GARCH_ged | 3 | 4.639826 |

The sigma forecast plot and the VaR backtesting plot for the MC-GARCH(1,1) model under the skewed Student's-t distribution are displayed below in Figures 7 and 8 respectively.

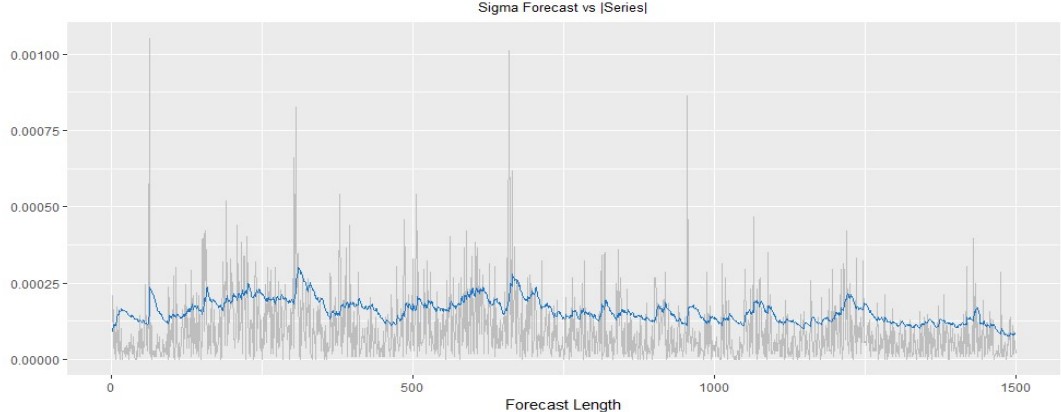

**Figure 7.** Sigma forecast plot.

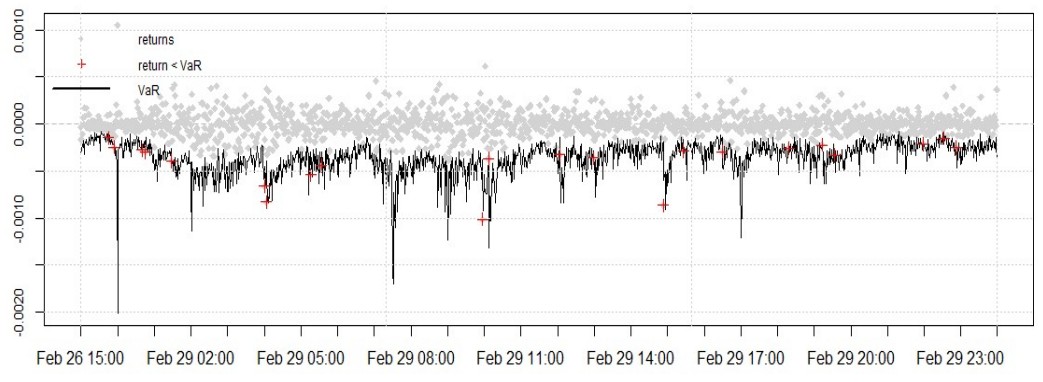

**Figure 8.** VaR backtest graph.

As noted in Singh et al. (2013), the spikes in the VaR forecasts as shown in the backtest plot in Figure 8 is due to the seasonal component during the opening of each trading day.

*4.8. Intraday ES Forecast*

The backtesting of the Expected Shortfall (ES) is now conducted, and three backtests are implemented to determine the accuracy of the ES forecast.

4.8.1. A Regression-Based ES Backtesting Procedure: the Bivariate ES Regression Backtest

The results for this backtest, both with and without bootstrapping, are shown in Table 8, below. All *p*-values are greater than the 5% significance level. The null hypothesis, which states that the 'ES forecasts are correctly specified' is not rejected. Moreover, the bootstrap *p*-values are also highly significant. Therefore, it can be concluded that the MC-GARCH models are able to forecast accurately the risk measure ES.

**Table 8.** Intraday ES forecast: bivariate ESR backtest results.

| Model | *p*-Value | Boot *p*-Value |
|---|---|---|
| MC-GARCH_std | 0.806 | 0.580 |
| MC-GARCH_sstd | 0.763 | 0.527 |
| MC-GARCH_jsu | 0.755 | 0.492 |
| MC-GARCH_ged | 0.868 | 0.664 |

Two classical ES backtests are employed to determine which model delivers the best ES estimates.

### 4.8.2. Exceedance Residual (ER) Backtest

The Exceedance Residual (ER) backtest of McNeil and Frey (2000) is also employed in this paper. The corresponding results are displayed in Table 9, below.

**Table 9.** Intraday ES forecast: Exceedance Residual (ER) backtest.

| Model | Expected Exceedances | Actual Exceedances | *p*-Value |
|---|---|---|---|
| MC-GARCH_std | 15 | 21 | 0.1845 |
| MC-GARCH_sstd | 15 | 22 | 0.1322 |
| MC-GARCH_jsu | 15 | 21 | 0.1302 |
| MC-GARCH_ged | 15 | 20 | 0.1077 |

The null hypothesis, which states that "Mean of Excess Violations of VaR is Equal to zero", is not rejected, since all *p*-values are greater than 5% confidence level. Based on this backtesting procedure, it can therefore be ascertained that the MC-GARCH models succeed in accurately predicting the ES estimates. Although the actual ES exceedances are comparable across the four MC-GARCH specifications, it can be observed that the MC-GARCH model under the GED error ditribution yields the least exceedances.

### 4.8.3. V-Tests

The V-Test statistics backtesting procedure can be regarded more as a diagnostic tool than a formal statistical testing procedure, since there is no null hypothesis involved.

Table 10 displays the results for the $V_1$, $V_2$ and $V_3$ test statistics. The first observation is that the sign of the $V_1$, $V_2$ and $V_3$ test statistics are positive, thus implying that all the models are, on average, overestimating the ES risk measure. Moreover, since the magnitude of the values of the test statistics are very close to zero, it implies that the models are only slightly overestimating the ES. These results speak in favour of the MC-GARCH models since risk managers are less concerned about overestimation of the risk metric as compared to an underestimation.

**Table 10.** Intraday ES Forecast: V-tests Backtesting Results.

| Model | $V_1$ | $V_2$ | $V$ |
|---|---|---|---|
| MC-GARCH_std | 0.0004419 | 0.0015778 | 0.0010099 |
| MC-GARCH_sstd | 0.0004391 | 0.0015690 | 0.0010041 |
| MC-GARCH_jsu | 0.0004383 | 0.0015667 | 0.0010025 |
| MC-GARCH_ged | 0.0004243 | 0.0015206 | 0.0009724 |

Furthermore, it can be observed that the magnitude of the $V_1$ test statistic is smaller for the MC-GARCH model under GED innovation assumption as compared to the other innovation assumptions thereby indicating that it performs relatively better. The same observation can be made for the other two test statistics, $V_2$ and $V_3$. Therefore, the MC-GARCH model under the GED error ditribution is the best model for ES under this backtest.

Figure 9 displays the ES forecasts for the MC-GARCH model under the GED innovation process. Once more it can be seen that the MC-GARCH models are able to adequately forecast ES.

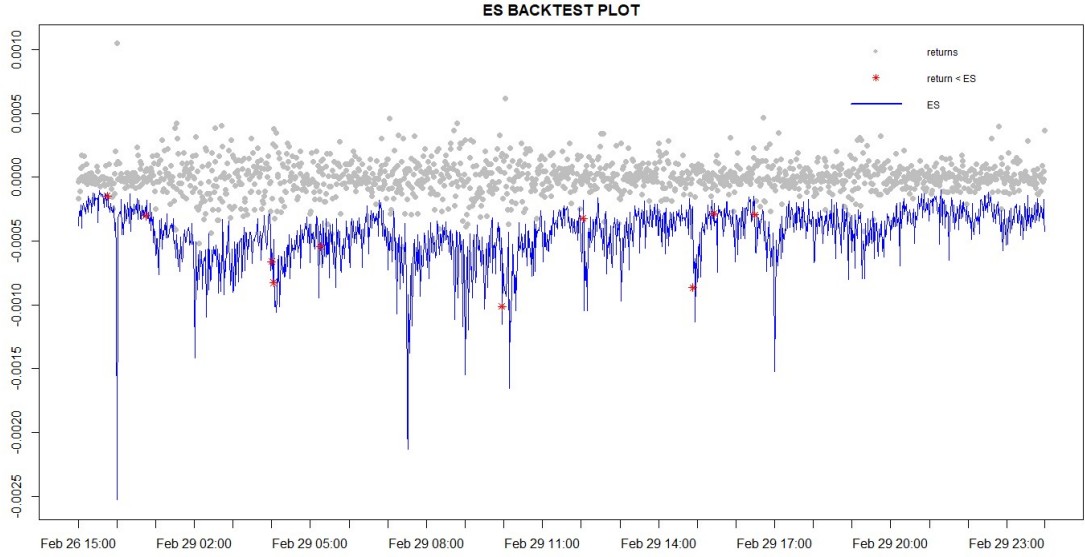

**Figure 9.** ES backtest graph for MC-GARCH_GED.

Figure 10 displays the forecast for both VaR and ES at 99% for the MC-GARCH_GED model.

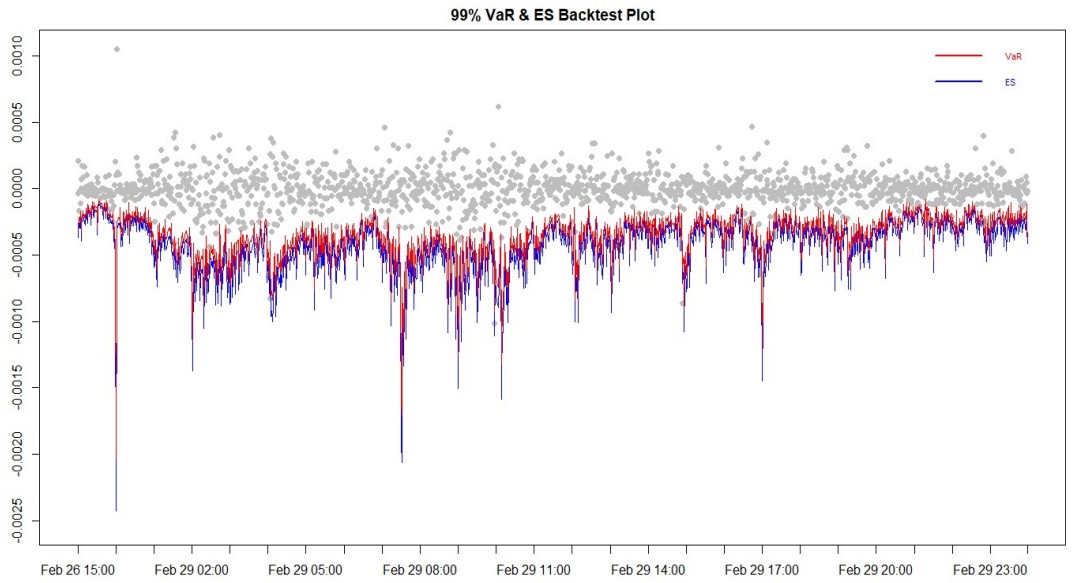

**Figure 10.** Backtest graph for MC-GARCH_GED.

## 5. Conclusions

A typical question that sparks a lot of interest in the high-frequency trading literature concerns which GARCH model tends to be the best when it comes to forecasting intradaily risk metrics such as Value-at-Risk (VaR) and Expected Shortfall (ES). This paper therefore focuses on the performance analysis of the MC-GARCH model in forecasting 1-min VaR and 1-min ES.

The first objective of this study was to determine which GARCH-type model gives the best in-sample fit to the daily EUR/USD returns for the daily variance forecast for the MC-GARCH model. It was found that overall the EGARCH(1,1) models were preferred over the GARCH(1,1) models. The EGARCH model under the GED innovation assumption however yielded the best results.

The second aim of the study was to analyse the effects of different distributional assumption for the innovation process of the GARCH models for both model fitting and forecasting. Overall, it was found that non-normal distributional assumptions gave better results for model fitting as well as

forecasting. This is due to the fact that non-normal distributions are able to take into account features such as excess kurtosis and asymmetry of the high-frequency EUR/USD returns. Furthermore, they are able to replicate these features when forecasting volatility. The MC-GARCH(1,1) model under the GED innovation assumption actually gave the best fit to the intraday data as per the ranking procedure carried out based on AIC, BIC and log-likelihood criteria.

The one-day ahead 99% intraday VaR values were forecasted using the MC-GARCH models. Three VaR backtesting procedures were carried out namely the Kupiec's test, the VaR duration-based backtest and a backtest based on an asymmetric VaR loss function. Based on the number of VaR violations, the MC-GARCH(1,1) model under the GED distribution gave the best results. When the asymmetric VaR loss function, which is a more robust backtesting procedure, was implemented, the MC-GARCH(1,1) model under the skewed Student's-t distribution minimised the loss function with the smallest value and proved to be the best model.

The one-day ahead 99% intraday ES was also forecasted using these models. Three backtesting procedures were employed for the ES, namely, the Bivariate ES regression backtest, the Exceedance Residuals backtest and the V-tests. It was found that the MC-GARCH models under the non-normal distribution assumptions are able to produce accurate intraday ES forecasts. The MC-GARCH(1,1) model under the GED distribution however yields the best results.

*5.1. Recommendations for Practitioners*

It is recommended to avoid the use of normal innovation distribution for MC-GARCH modelling, as it significantly overestimates risk. Such risk overestimation in the insurance and banking industries may actually lead to an excess of capital requirements, which may be unnecessary and hence loss-making for the institution. The MC-GARCH, under other innovation distributions such as Student's-t, the Skewed Student's-t distribution (sstd), the reparametrised Johnson SU (JSU) and the Generalised Error Distribution (GED), also overestimates the risk metrics, but yields empirical sizes closer to the expected size for both the VaR and ES. It is, however, recommended to employ the MC-GARCH(1,1) model under the GED distribution as it yields least overestimation results, which minimise the excess of capital requirement.

*5.2. Further Studies*

There still exist a multitude of areas for further research using the MC-GARCH models. Other distributional assumptions for the innovation process such as the skewed GED, Normal Inverse Gaussian (NIG) can be implemented. Moreover, the performance of the MC-GARCH models in predicting the risk metrics VaR and ES at higher confidence levels such as 99.5% or even 99.9% can also be assessed. The combination of Extreme Value Theory (EVT) with the MC-GARCH model can be analysed in forecasting intraday VaR and ES. A comparison between the MC-GARCH-EVT and the MC-GARCH models in predicting VaR and ES at different sampling frequency such as 1-min, 5-min and 10 min returns would be a particularly interesting study.

**Author Contributions:** Conceptualization, J.N.; Formal analysis, R.S.S.; Methodology, R.S.S. and J.N.; Software, R.S.S.; Supervision, J.N.; Validation, R.S.S.; Writing—original draft, R.S.S.; Writing—review & editing, J.N.

**Funding:** This research received no external funding.

**Acknowledgments:** The authors thank the reviewers for their valuable suggestions.

**Conflicts of Interest:** The authors declare no conflict of interest.

**Appendix A**

The preliminary analysis is laid out in this section.

It can be seen in Table A1 that the mean for both series hovers around 0 which in fact coincides with past studies on high-frequency financial returns. Moreover, the skewness values for both return series are negative which may imply that these series experience more negative shocks than positive

shocks and that there is a higher probability of obtaining a negative return. All kurtosis values being greater than 3, which is the kurtosis of any univariate normal distribution, imply that return distributions have thicker tails and sharper peaks at the centre as compared to a normal distribution. When comparing the degree of kurtosis and skewness for the 1-min returns (18.31108, −0.38839) with that of the daily returns (4.90965, −0.08208), it can be established that the kurtosis and the skewness values are much higher for the 1-min returns. This suggests that both the kurtosis and the degree of skewness increase with the frequency at which the data is recorded, thus confirming the findings of Andersen and Bollerslev (1998). The high kurtosis value for the 1-min returns is yet another stylised fact of high-frequency financial returns. The minimum value for the daily return occurred during the global financial crisis.

**Table A1.** EUR/USD Returns descriptive statistics.

|  | **1-min Returns** | **Daily Returns** |
|---|---|---|
| Mean | $1.23 \times 10^{-7}$ | $4.62 \times 10^{-5}$ |
| Standard deviation | 0.00017 | 0.00633 |
| Maximum | 0.00193 | 0.02781 |
| Minimum | −0.00332 | −0.03733 |
| Skewness | −0.38839 | −0.08208 |
| Kurtosis | 18.31108 | 4.90965 |
| Observations | 28,289 | 3159 |

These will aid to further confirm the presence of different stylised facts present in the return series.

To further demonstrate that both returns series deviate from normality, the Jarque-Bera (JB) test is carried out and their kernel estimates of the density are inspected. The results are presented in Table A2.

**Table A2.** Jarque Bera test.

|  | **Test Statistic** | ***p*-Value** | **Decision** |
|---|---|---|---|
| 1-min returns | 277,030 | 0 | Reject $H_0$ |
| Daily returns | 483.55 | 0 | Reject $H_0$ |

The *p*-value for the 1-min returns and for the daily returns series being equal to 0 for the JB normality test allows us to safely conclude, at a 5% significance level, that indeed these distributions do not follow a normal distribution. The same conclusion can be derived when analyzing the kernel density estimates in Figure A1 for both the 1 min returns (left) and the daily returns (right) since they clearly display leptokurticity.

To determine whether the series is stationary, the Augmented Dickey-Fuller (ADF) test is carried out on both return series. If the ADF test detects the presence of a unit root in the series, it can be deduced that the series is non-stationary and need differencing. Table A3 shows the results for the ADF test:

**Table A3.** ADF test.

|  | **Test Statistic** | **Lag Order** | **p-Value** |
|---|---|---|---|
| 1-min returns | −30.596 | 30 | 0.01 |
| Daily returns | −14.03 | 14 | 0.01 |

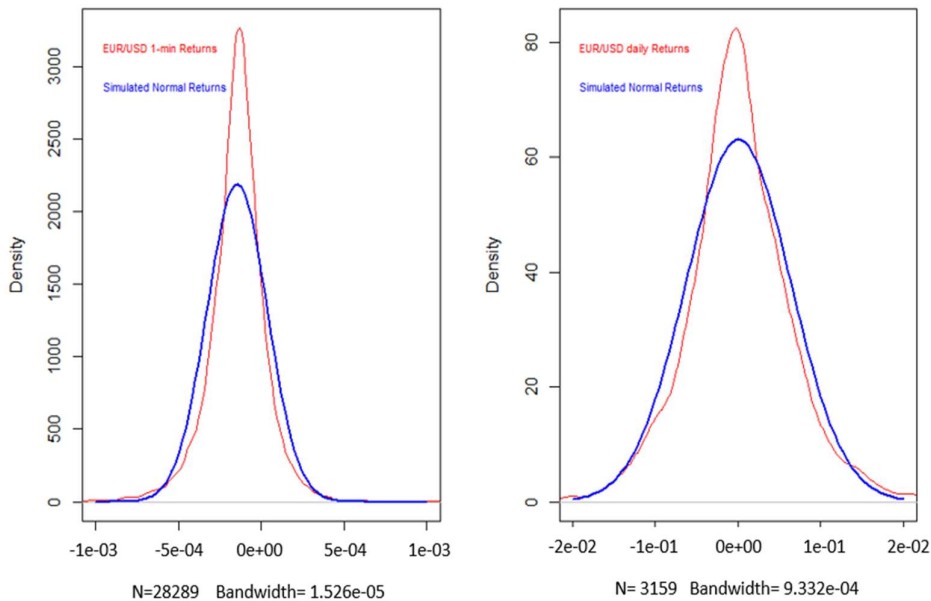

**Figure A1.** Kernel density estimates for 1 min returns (**left**) and daily returns (**right**).

The *p*-values for both returns series are less than 5% and this allows the rejection of the null hypothesis that a unit root is present in the series. Therefore, both return series are stationary and the order of the parameter *d* in the ARIMA($p, d, q$) model for both series is equal to 0.

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
