# Peer review of "Risk Model Validation: An Intraday VaR and ES Approach Using the Multiplicative Component GARCH"

_risks, doi:10.3390/risks7010010_

Round 1

Reviewer 1 Report

Please see the attached

Author Response

Rebuttal letter attached.

All comments from 3 reviewers were addressed.

All comments of Reviewer are addressed. Authors made used Microsoft Word's built-in track changes function to highlight any changes we made. 

Reviewer 2 Report

The paper is written clearly and it covers an interesting topic. However, some improvements should be done before it could be published:

- the novelty of the research and the contribution compared to the existing research should be more emphasized in the introduction

- financial time series specific characteristics are widely known for many decades now. Thus the section 2 should be trimmed down significanlty and the section 2 should be moved to section 4 where the results are presented. Section 2 should be renamed 4.1. data description (or similar)

- methodology section is too wide, researchers within this area know the distributions, the calculation of Var, ES, etc. Only newer things such as the model of Engle and Sokalska (2011) should be presented with more details. Thus, section 3 should be trimmed down as well.

- Instead of section 2 which is now the data, authors should make this into the section Previous literature (or similar title), with research relatable to this one and in presenting the results in this paper, comparisons should be made.

- policy recommendations (and for investors, etc.) are poor and should be extended upon. E.g. include a section prior to the conclusion section named Discussion. How can investors benefit from the results, etc. because now this just seems as an excercise of statistical interpretations

Author Response

Rebuttal letter attached.

All comments from 3 reviewers were addressed.

All comments of Reviewer are addressed. Authors made used Microsoft Word's built-in track changes function to highlight any changes we make. 

Reviewer 3 Report

REVIEW COMMENTS

Title: Risk model validation for High-Frequency exchange rate: an intraday VaR and ES approach using the Multiplicative Component GARCH.

Abstract

In this paper, the authors employ 99% intraday value-at-risk (VaR) and intraday expected shortfall (ES) as risk metrics to assess the competency of the Multiplicative Component Generalized Autoregressive Heteroskedasticity (MC-GARCH) models based on the 1-minute EUR/USD exchange rate returns. The authors test for Five distributional assumptions for the innovation process are used to analyse their effects on the modelling and forecasting performance. The authors used an MC-GARCH (1,1) model under the GED innovation assumption gave the best fit to the intraday data and gave the best results for the ES forecasts. However, the asymmetric Skewed Students-t distribution for the innovation process provided the best results for the VaR forecasts. The authors claim that this paper presents the first application (to the best of the authors’ knowledge) of the MC-GARCH model in forecasting the intraday Expected Shortfall (ES) under different distributional assumptions.

Comments:

The paper reads ok, but the title is too long, and should shorten it and provide more explanations in the introduction or abstract. A shorter title with clear communication is better. The review framework is well presented, and the results are good. The following remarks should be considered and addressed before a publication recommendation can be made.

1) The paper treats an interesting topic but does not have a good structure of flow for most manuscripts such as abstract, introduction, literature, problem statement, methodology, results and analyses, and then conclusion. I believe a well-structured outline will limit trying to make sense out of the paper.

2) Reducing redundant words is a practice that even seasoned writers get entangled in. I will recommend that the authors proof read the paper again to make it more concise.

3) If the problem statement is not being presented exclusively, then there is the need for the authors to outline clearly in the introduction, what the problem is and how they intend to solve it.

4) Some papers have considered other risks incorporating switching and how different is this paper from the others? What is novel about this article such that it should be published in this journal? The authors should state this in bullet points (between 3 and 5) in the introduction.

5) What are some of the managerial implications of this study in relation to the banking and insurance industry, and how can the study be generalized to fit other settings? Are the results exclusive? Or generic? A good indication of this can be done in the introduction.

6) In the introduction, the contribution to the literature is missing in terms of identifying the gaps and what you did to address that gap. Since there is no literature review, a comprehensive introduction is expected but the authors failed to address that. 

7) Overall, I think this is a good paper but there is the need for a thorough revision, especially in the contributions, to list the reasons why this paper should be accepted. I will encourage the authors to indicate at the end of the introduction, a summary or a pitch of the contributions. Also, at the end of the literature review, we should see the gaps in the literature that this paper addresses. Once these changes are made, the paper can be accepted.

Author Response

(The authors gave the same response as above.)

Round 2

Reviewer 1 Report

Please see the attached

Author Response

Dear Reviewer,

Thank you for your comments. We have addressed to them as follows:

- Regarding the presentation of the GARCH models, we have tried to present in a consistent and coherent way. For example, we have changed the notations of the daily variance component in the daily models (GARCH and EGARCH) to h_t instead of sigma^2_t. Moreover, in the MC-GARCH, we change omega, alpha_1 and beta_1 to omega^*, alpha_1^*, beta_1^* respectively. This is to contrast the GARCH used to model the intraday variance component with the GARCH used to model the daily variance component.

- The word "log return" is now used instead of "continuously compounded log return".

- Explanation on how to VaR and ES are calculated is included. We actually removed the latter explanation following a comment from another reviewer. We however concur with your comment. We have paid special attention to the notations used as well as regarding the definition of F in the calculation of VaR and ES.

Thank You.

Reviewer 2 Report

issues have been addressed, so the paper is now publishable 

Author Response

Thank you for your comments.